# Choroidal Vasculature Changes in Age-Related Macular Degeneration: From a Molecular to a Clinical Perspective

**DOI:** 10.3390/ijms231912010

**Published:** 2022-10-09

**Authors:** Serena Fragiotta, Luca Scuderi, Clemente Maria Iodice, Daria Rullo, Mariachiara Di Pippo, Elisa Maugliani, Solmaz Abdolrahimzadeh

**Affiliations:** 1Department of Neurosciences, Mental Health and Sense Organs (NESMOS), Ophthalmology Unit, St. Andrea Hospital, University of Rome “La Sapienza”, 00189 Rome, Italy; 2Department of Sense Organs, University Sapienza of Rome, 00161 Rome, Italy

**Keywords:** choroid, retina, age-related macular degeneration, choroidal vascularity index, aging

## Abstract

The contribution of choroidal vasculature to the pathogenesis of age-related macular degeneration (AMD) has been long debated. The present narrative review aims to discuss the primary molecular and choroidal structural changes occurring with aging and AMD with a brief overview of the principal multimodal imaging modalities and techniques that enable the optimal in vivo visualization of choroidal modifications. The molecular aspects that target the choroid in AMD mainly involve human leukocyte antigen (HLA) expression, complement dysregulation, leukocyte interaction at Bruch’s membrane, and mast cell infiltration of the choroid. A mechanistic link between high-risk genetic loci for AMD and mast cell recruitment has also been recently demonstrated. Recent advances in multimodal imaging allow more detailed visualization of choroidal structure, identifying alterations that may expand our comprehension of aging and AMD development.

## 1. Introduction

Choriocapillaris (CC) density decreases significantly with aging and may have a systemic basis [1]. Reduced perfusion of the CC, in combination with changes in the thickness and composition of Bruch’s membrane (BrM), further aggravates the metabolic homeostasis of the retinal pigment epithelium (RPE) and thus the photoreceptor cells [2,3,4]. Indeed, the age-related modifications of the choroidal vasculature have been reported to be associated with a wide range of BrM alterations, basal linear and basal laminar deposits, and RPE changes [4,5]. Based on clinical and histopathological evidence, two main theories on age-related macular degeneration (AMD) have been postulated. One theory suggests that the primary insult occurs at the RPE level, leading to secondary changes in the choroidal vasculature, supported by the evidence that RPE can secrete growth factors acting on the vasculature. A second theory is based on the hypothesis that impaired choroidal perfusion may be directly responsible for RPE dysfunction in AMD [6,7]. The close relationship between viable RPE and choroidal vessels was shown on histological models, demonstrating that decreased CC density is evident in eyes with geographic atrophy (GA) [1,6].

The refinement of choroidal imaging and the visualization and quantification of choroidal perfusion with incredible in vivo definition have greatly expanded the comprehension of choroidal modifications [8]. However, the intricacy of the molecular and structural aspects of aging need to be interpolated with the advances in choroidal imaging to fully understand the complex alterations involving RPE, BrM, and CC in AMD. Therefore, the present review explores the molecular, anatomical, and structural changes in the choroidal vasculature in aging and AMD and provides an excursus on in vivo imaging modalities.

## 2. Choroidal Vasculature

### 2.1. Vascular Anatomy of the Choroid

The arterial supply of the choroid originates from the posterior ciliary arteries (PCAs). One to five PCAs arise from the ophthalmic arteries, which are in turn branches of the internal carotid artery [9]. From their origin, the PCAs divide into several branches before piercing the sclera. Long posterior ciliary arteries, one or two in number, supply up to two-thirds of the choroid, ciliary muscle, and the major arterial circle. Short posterior ciliary arteries vary from 6 to 12, and the perpendicular terminal arterioles supply the CC, which irrorates BrM and the outer retina [10,11].

The arterial supply of the choroid is segmental from the PCAs to the CC [11]. The segmental and lobular organization of choroidal vasculature can be appreciated at both the posterior pole and the equatorial retina [5].

### 2.2. Structure and Function of the Choroidal Vascular Layers

The vascular region of the choroid, from the innermost to the outermost layers, consists of the CC, Sattler, and Haller layers [12].

The CC represents the innermost choroidal layer constituted of a capillary plexus beneath BrM. The endothelial cells of the CC are fenestrated mostly on the retinal side, expressing vascular endothelial growth factors (VEGF)-1 and -2 receptors. The CC also expresses intracellular adhesion molecule-1 (ICAM-1), responsible for leukocyte adhesion via CD11a/CD18 or CD11b/CD18 integrins on endothelial cells [13,14].

CC organization consists of a lobular pattern in which each lobule is supplied by a terminal choroidal arteriole with no functional anastomosis, thus creating watershed zones susceptible to hypoxia [11]. The structural and functional organization of the CC can be variable, and two main interpretations of the functional arrangement have been proposed: a lobule with a central venule surrounded by peripheral arterioles or a functional lobule with a central arteriole and peripheral venules. The regular lobule can be distinguished at the posterior pole, while the lobular arrangement becomes less dense toward the peripheral retina [5,9,11,15].

Under light conditions, the oxygen provided to photoreceptors comes from the CC. A lack of autoregulation of the choroid may profoundly affect the oxygen tension under certain conditions with irreversible hypoxic damage to the photoreceptors [16].

Sattler’s vascular layer comprises medium-to-small arteries, the arterioles feeding the capillary layer, and veins. Each arteriole supplies a hexagonal or lobular domain in the CC. Haller’s vascular layer contains large-diameter vessels of arteries and veins [12,17]. 

## 3. Molecular Aspects in Choroidal Vasculature

### 3.1. Genetic Factors Associated with Pathologic Choroid in AMD

Complement factor (*CFH)* rs1061170 risk genotype, a well-known single nucleotide polymorphism (SNP) in AMD, was associated with choroidal thinning (−14.7% with the risk allele C) in healthy elderly individuals, suggesting a common pathway involving choroidal thinning in AMD pathogenesis [18]. Similar results were reported by Mullins et al., who revealed that the eyes homozygous for the Y402H allele presented significant choroidal thinning, compared with those homozygous for the low-risk Y allele (−23.6% thinner). The authors hypothesized that the *CFH* polymorphisms associated with impaired protein function or localization could be associated with the increased formation of membrane attack complex (MAC) and further complement activation, leading to CC injury in early AMD [19].

Most studies investigating the association between genetic polymorphisms and choroidal vascular changes involved a specific subtype of neovascular AMD (nAMD), namely polypoidal choroidal vasculopathy (PCV). This phenotype represents almost 50% of nAMD cases in the Asian population and is characterized by nodular or aneurysmatic dilatations arising from the neovascular network [20]. The I62V polymorphism in the *CFH* gene contributed to subfoveal choroidal thickening in eyes with PCV, while the Y402H polymorphism in the *CFH* gene and the A69S polymorphism in the age-related maculopathy susceptibility2 (*ARMS2*) gene were not associated with choroidal thickness. Since the *CFH* gene is a key regulator of the alternative pathway of the complement system, the authors speculated a possible link between inflammation and choroidal modifications in PCV eyes [21]. Another study on eyes with PCV reporting the factors associated with subfoveal choroidal thickening included younger age, shorter axial length, G-allele frequency in *ARMS2* A69S (rs10490924), and T-allele frequency in *CFH* (rs1329428) [22].

### 3.2. Molecular Properties of the Choroidal Vascular Layer

Human leukocyte antigen (HLA) expression has been demonstrated in both healthy and AMD eyes and is mainly expressed on the endothelial cells of the CC. An increased HLA class II immunoreactivity was also observed in AMD eyes during drusen formation [23,24]. The association of HLA antigens in AMD suggests a pathogenetic role in driving the cellular immune-mediated response against local antigens or newly presenting antigens, especially drusen and AMD-related evolutive changes [23].

As mentioned above, ICAM-1 is constitutively expressed in the endothelial cells of the CC in healthy subjects. Its expression is associated with leukocyte recruitment in the choroid, thus expanding the immune response [14,25].

The endothelial cells of the CC are particularly dependent upon VEGF, which is crucial for CC development and maintenance [26,27,28,29]. VEGF secretion is also essential in maintaining the endothelial CC fenestrations, and VEGFR-2 is actively involved in CC development and maturation during gestational age [30].

Mast cells play crucial roles in inflammation and are involved in several pathophysiological processes, including wound healing, tissue regeneration and remodeling after injury, and angiogenesis [31]. Mast cells and macrophages are the main leukocytes detectable in the choroid and proved to be involved in the pathogenesis of AMD [32].

The complement system is a molecular cascade part of the innate immune system that drives the inflammatory response, cooperates with antibodies to destroy pathogens, and improves the individual immune response. When activated, the complement cascade proteases cleave specific proteins, releasing the cytokines that further activate and amplify the cascade. These events initiate the cell-killing membrane attack complex (MAC) that perforates cellular membranes. The complement system consists of three distinctive pathways: the classical, the alternative, and the mannose-binding lectin pathway. The classical pathway is triggered by the antigen–antibody complexes that bind the C1 complex. The alternative and mannose-binding lectin pathways are activated without needing any antigen–antibody interaction. The complement system is constantly activated at low levels in normal eyes under the control of the complement regulatory proteins that maintain a balanced activation to avoid eye damage [13].

### 3.3. Age-Related and Pathologic Changes

The interaction between leukocytes and BrM is mainly related to the topographical distribution of drusen and basal deposits. Collagenous debris is distributed within and beneath BrM, representing a target of phagocytic activity. In the early stages of a disease, the cellular interactions are limited to the BrM, whereas at later stages, the cells can be visible on the inner retinal surface through breaks of the BrM. These breaks may favor the passage of neovessels with a possible role in the choroidal neovascularization progression [33].

Mast cells are resident choroidal inflammatory cells that tend to increase in all the choroidal areas of eyes with early AMD. Mast cells appeared degranulated and located close to the CC or in spaces where the CC had degenerated. An increasing number of degranulated mast cells were also evident in the paramacular choroidal locations of eyes with exudative AMD and GA. The degranulated mast cells in neovascular AMD were also located adjacent to the feeder vessels passing through BrM breaks, in the basal laminar deposits overlying neovascularization, or within lesions. Moreover, the mast cells were primarily located in the Sattler’s layer and CC areas with complete RPE atrophy in GA eyes [31]. In eyes with high-risk genetic loci on chromosomes 1 and 10, increased mast cell proteases and choroidal mast cells were evident in the absence of clinical AMD changes and after AMD had developed. These findings suggested that mast cell infiltration and degranulation are early events in AMD pathogenesis. Furthermore, the mast-cell-protease-mediated degradation of the CC basement membrane leads to endothelial damage. However, the altered integrity of BrM also facilitates the access of large blood-derived proteins between the RPE–basal lamina and the inner collagenous layers, causing drusen formation and progressive RPE damage [34].

Resident choroidal macrophages are absent in normal human eyes without inducible nitric oxide synthase (iNOS) expression. In eyes with early AMD (group III-IV) with pigmentary changes, macrophages occupied choroidal capillaries or the intercapillary pillars. In this context, the choroidal vascular endothelial cells and perivascular cells also expressed iNOS. In the same group, the macrophages were initially located in correspondence with soft drusen, but they tended to increase in the presence of basal laminar deposits, being recognizable in all the examined eyes as the deposits became thicker. When choroidal neovascularization occurred, the macrophages further increased in number and frequently eroded the intercapillary pillars of BrM. In the context of GA, the macrophages were reduced underneath the atrophic area but were mainly distributed at the atrophic border and adjacent to viable RPE. In disciform scarring, the macrophages were detectable within the inactive and active scar, while iNOS expression was only appreciated within the active lesions [35].

The dysregulation of complement activity with the evidence of C3 and C5 complement fragments and MAC C5b-9 have been identified in the context of drusen and choroidal capillary pillars, as well as in the vitreous of AMD eyes [13,36,37]. The common genetic polymorphisms carrying an increased risk for AMD involve the regulation of the complement pathway. A polymorphism (Tyr402His) in the CFH gene, which mainly controls the alternative complement activation, reduces the affinity for C-reactive protein (CRP), increasing the levels of unbound CRP in the choroid, leading to uncontrolled chronic inflammation [37,38,39].

## 4. Structural Changes in Choroidal Vasculature with Aging and Disease

### 4.1. Age-Related Changes in the Choroidal Vascular Layer

The spectrum of age-related choroidal changes, identified by Green and Key, was characterized by a sclerosed CC with the hyalinization of the intercapillary septae and choroidal thinning clinically visible with evident choroidal vessels and a tigroid fundus. A variety of BrM changes were also evident, including thickening, hyalinization, calcification, and breaks. Despite this, the histopathological evaluation of the CC may be complicated without showing a consistent pattern of CC loss, thus leaving the issue of the primary disease locus still open [4].

The histological sections obtained through the regions with CC dropout revealed a wide range of BrM changes, including delamination, cracks, discontinuities, and basal linear and basal laminar deposits. When the CC dropout was significantly accompanied by basal linear and basal laminar deposits, many variations of choroidal neovascularization formations occurred. These included sub-RPE monolayers of the endothelium, viable vascular networks within basal linear deposits, and atrophic networks within basal linear deposits or delaminated BrM [5].

Choroidal blood volume represents the amount of blood measured at a given point through laser Doppler flowmetry. This parameter presented a significant inverse relationship with age, demonstrating a reduced volume with aging. When dividing the population into two groups according to age, the older group showed a reduction of about 29% in both choroidal volume and blood flow compared with younger patients. The decreased choroidal flow is mainly related to a decrease in volume, supporting the idea that choroidal changes may play a role in the pathogenesis of AMD [40]. 

### 4.2. Subretinal Drusenoid Deposits (Reticular Pseudodrusen) and Choroidal Hypoxia

The structural lobular organization of the choroid has been considered responsible for increased susceptibility to ischemia in the watershed zones located at the edges of perfusion areas. Eyes with reticular pseudodrusen (RPDs)/subretinal drusenoid deposits (SDDs) exhibited well-demarcated watershed zones in 83.3% compared with 40% of healthy controls. In 88% of cases, the RPDs/SDDs were located partly or entirely within the watershed zones [41].

The vascular changes associated with SDDs and basal linear deposits (BLinDs) included the absence of endothelial cells in an arch-like pattern delimited by intercapillary pillars, reaching 17.3% when considering either lesion. However, the choroidal alterations were also visible outside the areas occupied by SDDs/BLinDs, comprising choroidal thinning, significant vessel loss, and stromal hyalinization in the macular region [42].

### 4.3. Choroidal Changes in Age-Related Macular Degeneration

AMD is a multifactorial and complex disease that includes several possible mechanisms: genetic risk factors encoding for the complement system such as complement factor H, CFHR, C2, C3, CFI and SERPING1, inflammatory, oxidative, lipidic, mitochondrial, and microvascular factors [17,43]. Evidence of CC impairment in AMD emerged from several studies with different approaches, including prolonged choroidal filling on both indocyanine green and fluorescein angiography in early AMD, Doppler flow studies, a proteomic analysis that found the loss of CC protein (CA4 and HLA-A) with the preservation of RPE proteins (CRALBP and RPE 65), histopathological studies, and more recently, with the advancements in multimodal imaging [43,44,45,46,47,48,49].

Drusen formation appears to follow the distribution of underlying choroid. In the early stages of drusen formation, these tended to distribute in the stroma between the capillaries associated with venules. BrM modifications emerged in later stages [50]. The spatial association between drusen and CC was reported by various authors. Sarks et al. [51] noted that early preclinical drusen tended to accumulate over CC pillars, with the hyalinization reaching the capillaries that appeared widely spaced when hyalinized. Lengyel et al. [52] also confirmed the spatial distribution of drusen consistently located internal to the intercapillary pillars of the CC. Further, CC density was found to be associated with the extent of drusen and sub-RPE deposits, suggesting that the loss of endothelial cells promotes drusen formation [53]. CC attenuation in early AMD is localized in the submacular choroid, leading, in some cases, to incipient neovascular buds at the border of the area affected [54]. CC narrowing has been often described in AMD, with a progressive capillary decline from 71% in group I (small drusen) to 44% in group IV (incipient atrophy) [55].

In eyes with GA, RPE atrophy was usually accompanied by CC loss, with some degrees of surviving CC where the capillaries were highly constricted [6]. Once the capillaries lost their fenestrations, they were enveloped by fibroblast processes and collagens and retracted from BrM with increasing spacing [55]. Furthermore, the surviving RPE cells associated with residual CC demonstrated the presence of neovascular membranes, supporting the role of RPE cells in providing a stimulus for neovessel formation and stabilization [4,6]. In eyes with GA, the CC density was 54% that of normal eyes. A decreased CC density was also appreciated accompanying basal laminar deposits with a reduction of 63% and a decrease of 43% in those areas with disciform scars [1].

The choroidal blood flow parameters obtained through laser Doppler flowmetry, including choroidal volume, flow, and velocity, demonstrated a progressive decline with increasing AMD severity [56]. In the eyes that developed macular neovascularization, choroidal volume and flow were significantly reduced [57].

A comprehensive model embracing genetics, complement, microvascular alteration, and drusen formation has been proposed. In this model, the complement activation, more pronounced in eyes with high-risk genotypes, develops within the CC with consequent VEGF expression in the RPE and ICAM1 in the CC that promotes angiogenesis and leukocyte recruitment. The membrane attack complex (MAC) generation leads to a progressive CC loss. Moreover, CC impairment ends in a failure to remove debris from the RPE/Bruch’s membrane, leading to drusen development along with retinal and RPE progressive hypoxia [43].

Despite this, other studies indicated a well-preserved CC in AMD, hypothesizing that basal laminar/basal linear deposits may alter the oxygen diffusion between RPE and CC. Therefore, hypoxic RPE and the outer retinal layers may produce angiogenic factors that can act on CC. According to this, the choroidal vascular changes in AMD appeared to be secondary to RPE and outer retina damage [4,58].

The controversy on the site of the primary insult in AMD, CC, or RPE, still remains. However, an alternative hypothesis was formulated by the Lutty group (unpublished data), who proposed that the RPE degenerates first in GA, while the CC is primarily affected in neovascular AMD, especially adjacent to the neovascularization [17]. 

## 5. Choroidal Imaging and Assessment In Vivo

Recent advances in multimodal imaging have enabled a deepened study of choroidal structure, identifying the potential changes within the choroid and CC of AMD eyes in vivo and developing valuable biomarkers of disease stage and phenotype [59]. Choroidal vasculature layer visualization in vivo is considered particularly important to understand the pathophysiological changes occurring in AMD. It is crucial to identify the structural microvascular changes preceding macular complications and is pivotal for the diagnosis of neovascular AMD. Until recently, the visualization and study of the choroid were only limited to postmortem histology or the indirect observations made through invasive dye-based angiography [60]. However, the tissue fixation performed before histological analysis causes the shrinkage of vasculature and does not reflect physiological variations in the vascular lumen, providing only a rough estimate of the choroid in vivo [61].

Earlier studies showed slow choroidal filling in AMD eyes through dye tests, not necessarily colocalized with drusen, indicating that a continuous layer of debris in BrM may reduce the metabolic exchange between the RPE and the choroid [62,63]. The invasive dye tests were progressively replaced by non-invasive technologies providing an increasingly better definition of choroidal anatomical planes and structures [64]. Currently, commercially available instrumentations enable qualitative and quantitative assessments of choroidal vessels and the CC. The main imaging modalities available on the market to study the choroidal vascular layer and the validated analyses to reveal potential pathological modifications are summarized below. 

### 5.1. Enhanced Depth Imaging and Choroidal Thickness

#### 5.1.1. Instrumentation, Technique, and Limitations

In spectral-domain optical coherence tomography (OCT), the “zero-delay” line is a reference point representing the image focal point where image resolution is optimal. Standard software on instrumentation automatically sets the zero-delay line at the inner retina surface. Therefore, conventional SD-OCT devices present significant limitations in visualizing deeper structures such as the choroid [48,65]. With the introduction of enhanced-depth imaging (EDI) spectral-domain mode in 2008 by Spaide et al. [48], the choroid was enhanced using an inverted OCT B-scan, bringing the choroid closer to the zero-delay line to increase its visualization. EDI mode was then optimized using a built-in software modality that automatically shifts the zero-delay line on the choroid–scleral interface without the need to reverse the OCT B-scan [66,67]. 

The advent of swept-source OCT technology, using a longer wavelength of 1050 nm, demonstrated a better penetration of the OCT light probe with more accurate visualization of the choroidal structures [68,69]. An optimal distinction of the choroid–scleral junction enables a reliable calculation of choroidal thickness, which can be calculated by tracing a distance from the outer border of the RPE to the inner surface of the choroid–scleral junction using a digital caliper (Figure 1) [66,70,71,72,73,74,75]. Spectral-domain and swept-source technologies are generally comparable in terms of choroidal thickness variability, although the variability could increase under pathological conditions. This variability can be reduced by manually adjusting the segmentation boundaries to minimize the difference between the two OCT devices of ~50 μm [76,77].

Choroidal thickness seems to vary within the posterior pole, and the thickness underneath the fovea has been reported to be the thickest point in normal subjects [61]. The mean subfoveal thickness in normal subjects is estimated to be between 287 and 335 μm [61,78]. Several physiological variations in choroidal thickness have been reported including gender, age, and axial length [66,78,79,80,81,82]. The fluctuations in choroidal thickness may follow diurnal variations, involving mainly the luminal component with a greater thickening in the morning hours [83]. As mentioned above, the coexistence of SDDs and choroidal thinning may be influenced by several confounding factors that limit the ability to establish a robust causal relationship [84,85].

Another source of variability of choroidal thickness is represented by the repeatability between spectral-domain and swept-source devices for the choroidal thickness greater than 400 μm. In this condition, the intraclass correlation coefficient was significantly decreased with spectral-domain OCT limiting its accuracy but maintained using a swept-source device, which should be preferred in such cases [86].

#### 5.1.2. The Clinical Significance of Choroidal Thickness Variations in AMD

Several studies investigated potential choroidal thickness variations in different AMD stages. The underlying idea was that an age-related loss of choroidal thickness associated with vascular sheathing or obliteration induces inadequate support to the overlying RPE and outer retina preceding and contributing to AMD pathological changes [61,80,87]. Choroidal age-related thinning was well-recognized in the subfoveal area but also diffusely within the central 12 mm, with a decline of approximately 2 μm per year of age across all ethnic groups [66,81,88,89,90]. Choroidal thickness tended to reduce according to AMD severity, with Sattler’s layer thickness being markedly reduced in intermediate AMD and nAMD, compared with normal eyes [91]. The ratios of Sattler’s layer thickness and Haller’s layer thickness to the total choroidal thickness also tended to decrease with aging [89].

Despite the hypothesized choroidal thinning progressive changes from normal aging to AMD, the early stage of AMD (drusen < 125 μm in diameter) may have an unaffected choroidal thickness [92]. Lee et al. [93] distinguished the entity of subfoveal choroidal thickness variations according to extracellular deposit subtypes, demonstrating a thicker choroid in pachydrusen compared with soft drusen and SDDs, where choroidal thinning was obvious.

A thicker choroid associated with pachydrusen is not surprising considering the pathogenesis of the pachychoroid spectrum, where dilated outer choroidal/Haller’s layer vessels (pachyvessels) and thinning of the CC and Sattler’s layer represent distinctive features [94]. In the context of the pachychoroid spectrum, the most common complication is represented by macular neovascularization with a characteristic choroidal thickening, representing a distinctive biomarker in distinguishing this phenotype from AMD [95,96,97]. Pachyvessels were often identified underneath PCV, even for eyes with a subfoveal choroidal thickness of less than 200 μm. Although no univocal consensus exists on the definition of choroidal thickening, a cut-off of 200 μm was arbitrarily established according to the values obtained from a normal population [98]. Choroidal thickening can be focal or diffuse, with the dilation of choroidal vessels and inner choroidal attenuation beneath the neovascular lesion, delineating an impairment in the thickness ratio of the outer/inner choroid [94]. The dilation of choroidal vessels is recognizable in the entire pachychoroid spectrum, including pachychoroid pigment epitheliopathy, central serous chorioretinopathy, and PCV/aneurysmal type 1 lesions [99,100].

Choroidal thickness can be reduced in other forms of neovascularization, typically type 3 lesions [101]. Likewise, choroidal thinning is a characteristic in eyes with RPDs/SDDs and geographic atrophy [68,102,103,104]. Several in vivo studies confirmed a marked alteration in choroidal parameters in eyes with RPDs/SDDs, using choroidal thickness, choroidal vascular thickness, choroidal vascularity index (CVI), and optical coherence tomography angiography (OCTA) [105,106,107,108,109,110]. Although most of the existing literature reported a thinner choroid in eyes with SDDs, some studies did not observe choroidal thinning between AMD eyes and SDDs, suggesting a non-univocal association between SDDs and choroidal thickness [84,85].

### 5.2. Choroidal Vascularity Index (CVI)

#### 5.2.1. Technical Procedure for Obtaining CVI, Strengths, and Limitations

The choroidal vascularity index (CVI) is a ratio that defines the proportion of the vascular component of the choroid compared with the total choroidal area. This index can be obtained through post-processing using the open-source software ImageJ (distributed by Fiji, https://imagej.net/Fiji/Downloads, last accessed on 15 July 2022) for imaging analysis [49,111]. The post-processing method is usually applied on a single subfoveal OCT B-scan with optimal visualization of the choroid–scleral junction. However, a volumetric CVI estimation calculating several consecutive B-scans can also be obtained [44,112,113,114].

After the OCT B-scan is selected, the .tiff image is imported into Fiji software. A region of interest (ROI) is traced with the polygon tool, with the upper boundary at the level of the RPE and the lower boundary at the choroid/scleral junction. Three large choroidal vessels (>100 microns) are selected using the oval selection tool. Thus, a binarization is performed through Niblack’s autolocal threshold technique, to analyze the mean and standard deviation of dark and light pixels in the ROI [111]. The software analysis considers the areas of dark pixels as luminal areas (LAs), while the totality of the pixels in the ROI defines the total area (TA). The ratio between TA and LA is the formula applied to calculate CVI (Figure 2) [49,112,115].

Since its introduction, the CVI has demonstrated a more informative and accurate assessment of the choroidal vascular status allowing a deep understanding of choroidal changes in eye diseases far more accurately than choroidal thickness. The main disorders considered include inherited retinal disorders, inflammatory chorioretinal disorders, pachychoroid disease spectrum, AMD, myopia, diabetic retinopathy, glaucoma, ocular surgery, anterior ischemic optic neuropathy, and hemodialysis [49]. 

The CVI parameter is considered more sensitive than the choroidal thickness in revealing choroidal alterations, as it is less influenced by confounders. This index has a narrower range of variability across the normal population compared with the high variability exhibited by choroidal thickness. Further, choroidal thickness is closely associated with age, demonstrating a decreasing thickness with aging, while CVI remains constant [88,112]. The repeatability of CVI is more consistent than the choroidal thickness, with fewer fluctuations related to the subfields analyzed and the axial length [116]. The reproducibility of CVI was assessed using three different scanning patterns, including a subfoveal single EDI scan, the B-scans covering the central 1000 micron, and the total macular scans (31 B-scans) within a 30 × 25 degrees area centered on the fovea. The three scanning patterns demonstrated a high intraclass correlation coefficient (ICC), confirming the CVI reproducibility and indicating that a single subfoveal scan was representative of the entire posterior pole in healthy subjects [114]. The main disadvantage of using CVI consists of post-processing the images into external software for imaging elaboration. The lack of in-built software available on OCT devices may limit its utility in clinical practice, restricting its use for research purposes. Another point is related to the different algorithms for binarization used that showed poor agreement with no evidence of which algorithm is better for the CVI calculation [49,117].

#### 5.2.2. Clinical Interpretation of CVI in AMD

CVI was significantly impaired in AMD eyes with differences among the various phenotypes explored. In particular, CVI was altered considerably in eyes with pseudodrusen/SDDs and GA compared with the controls with a relevant impairment of the luminal component and global CVI, supporting progressive choroidal vascular degeneration with stromal replacement [108]. This is further supported by the evidence of a progressive CVI reduction accompanied by a constant increase in the stromal component in GA eyes over time (average of 18.3 months) [118]. The CVI calculated within the area of GA demonstrated an association with lesion enlargement, reflecting the possibility that a decreased choroidal perfusion contributes to CC dropout, leading to GA expansion [119].

CVI reduction in RPDs/SDDs was further confirmed in a recent study, where the eyes with drusen greater than 125 μm did not show significant changes compared with healthy controls [120]. CVI changes in AMD seem to follow a biphasic trend, with an initial increase during the earlier stages of AMD, followed by a progressive decrease in the advanced stages. The authors hypothesized that the initial phase might result from compensatory vascular engorgement due to the CC loss and inflammatory changes preceding choroidal degeneration [106].

### 5.3. Optical Coherence Tomography Angiography

#### 5.3.1. Choriocapillaris Flow Void Quantification

OCTA enables the visualization of the different choroidal vascular slabs, including the CC that appears as a granular structure. Deficits in CC have been characterized as flow voids starting with focal defects smaller than a choroidal globule that tend to increase with age, hypertension, and in eyes with pseudodrusen and late AMD [121]. The flow voids assessment is performed by importing the CC slab into a software program for image processing (Fiji). Once imported, the CC slab is thresholded using the Phansalkar method and then analyzed using the “Analyze Particles” tool to count the areas of flow voids, their number, and total and averaged area [46].

#### 5.3.2. Choriocapillaris Flow Voids in AMD

The increasing CC flow deficits were associated with complete RPE and outer retina atrophy progression in eyes with intermediate AMD [122]. Moreover, the presence of pseudodrusen was accompanied by a decreased flow at the CC level with increased flow deficits [123,124]. The automatized evaluation of flow voids using post-processing elaboration through dedicated imaging software can be challenging (Figure 3). In fact, the commonly used compensation technique for swept-source OCTA can create artifactual flow deficits in correspondence with hyperreflective areas on the OCT B-scan, leaving possible limitations of the methodology [125]. Despite this, the CC deficits revealed on OCTA can predict drusen growth and enlargement and RPE atrophy with GA expansion [46,126]. The highest percentage of flow voids has been found within 600 μm of the GA margins, but the CC flow deficits were associated with atrophy enlargement despite the localization, including regions outside of the GA [119]. Moreover, CC perfusion was also associated with retinal sensitivity under scotopic conditions in early and intermediate AMD, confirming the crucial role of CC even in the early stages of the disease in preserving the outer retina and highlighting the relevance of CC in AMD pathophysiology [47].

## 6. Conclusions

The present review offers a comprehensive overview of the primary molecular and choroidal structural changes occurring with aging and AMD. The complexity of AMD resides in its multifactorial pathogenesis involving genetics, inflammatory response, microvascular, and metabolic impairment, causing the accumulation of extracellular deposits. The component that contributes to starting the pathogenetic AMD process remains controversial, but choroidal involvement is undeniable. 

The molecular alterations targeting the choroid in AMD involve several inflammatory mechanisms spacing from the human leukocyte antigen (HLA), complement system, cytokines and growth factors, leukocyte interaction at the BrM level, and the mast cell and macrophage infiltration of the choroid. Genetic predisposition with high-risk genetic loci for AMD can result in complement activation and increase mast cell infiltration and degranulation. In several histopathological studies, the CC was attenuated from the early stages of drusen formation but this was not univocal, as few studies demonstrated a well-preserved CC, hypothesizing that basal laminar/basal linear deposits may alter oxygen diffusion between the RPE and CC. The involvement of choroidal vasculature in AMD is also highlighted by the recent advances in multimodal imaging that enable enhanced visualization of choroidal structures. The robustness of choroidal thickness as a biomarker has significantly been reconsidered, given the physiological variations, the potential confounders, the heterogeneous variability under pathological conditions, and thus, the inability to establish a univocal relation with AMD features. Despite this, sophistication in imaging post-processing allows a more accurate evaluation of choroidal perfusion using the choroidal vascularity index calculated on OCT B-scans. Using OCTA, the different choroidal vascular slabs are identifiable with the direct visualization of the CC and its progressive rarefaction with flow voids accompanying aging and AMD. Choroidal thinning, vascular dropout, and stromal hyalinization accompanying RDPs/SDDs in the macular region and seen on histopathological specimens were also confirmed in vivo by an evident choroidal thinning, CVI reduction, and CC flow voids on OCTA. In GA eyes, the progressive choroidal vascular degeneration associated with stromal replacement seen on histopathology can be appreciated in vivo by observing CVI modifications, which confirmed a reduction in the luminal component accompanied by a stromal increase with time.

Although these novel technologies enable the visualization of the choroidal ultrastructure with an impressive histological correspondence, the resolution and algorithms still need to be potentiated for an optimal assessment of the CC. Although the choroidal vascular layer in AMD presents undeniable structural alterations and modifications contributing to the progression of the disease, there is no sufficient evidence to consider the choroid as the primary contributor in AMD pathogenesis, where several factors are involved and not adequately explored. In this panorama, the intricate molecular mechanisms remain to be fully elucidated and hopefully interpolated with in vivo biomarkers.

Future studies would be desirable to characterize these aspects but also to provide automatized tools for choroidal evaluation available on commercial devices. The progressive improvement of imaging technologies and image manipulation with dedicated software continually expands the knowledge of aging and AMD development.

## Figures and Tables

**Figure 1 ijms-23-12010-f001:**
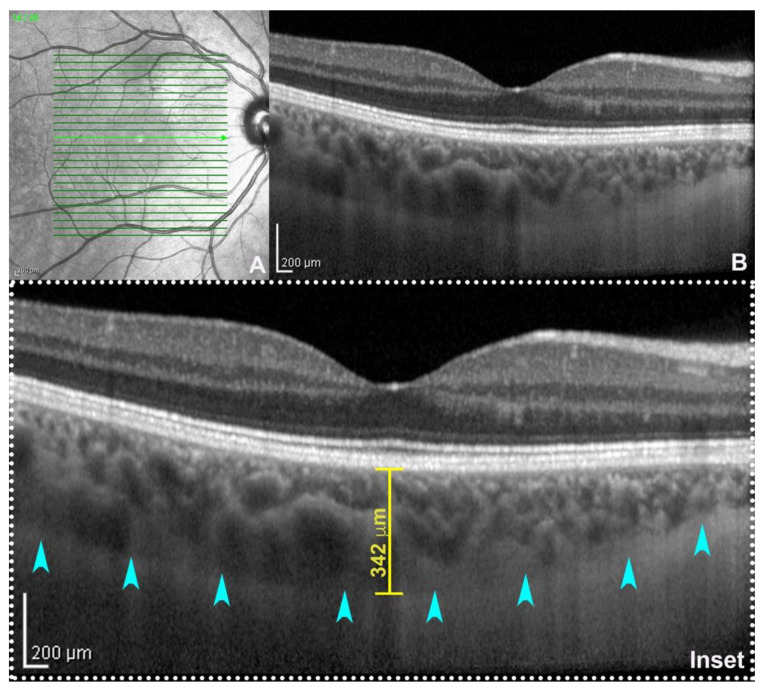
Enhanced depth imaging (EDI: (**A**) near-infrared reflectance; (**B**) spectral-domain optical coherence tomography (SD-OCT, Heidelberg Engineering, Germany) subfoveal B-scan acquired using EDI-mode. On magnification (inset), the choroid–scleral junction is clearly detectable (teal arrowheads) allowing the calculation of subfoveal choroidal thickness traced between the outer border of the retinal pigment epithelium and the inner surface of the choroid–scleral junction through a digital caliper.

**Figure 2 ijms-23-12010-f002:**
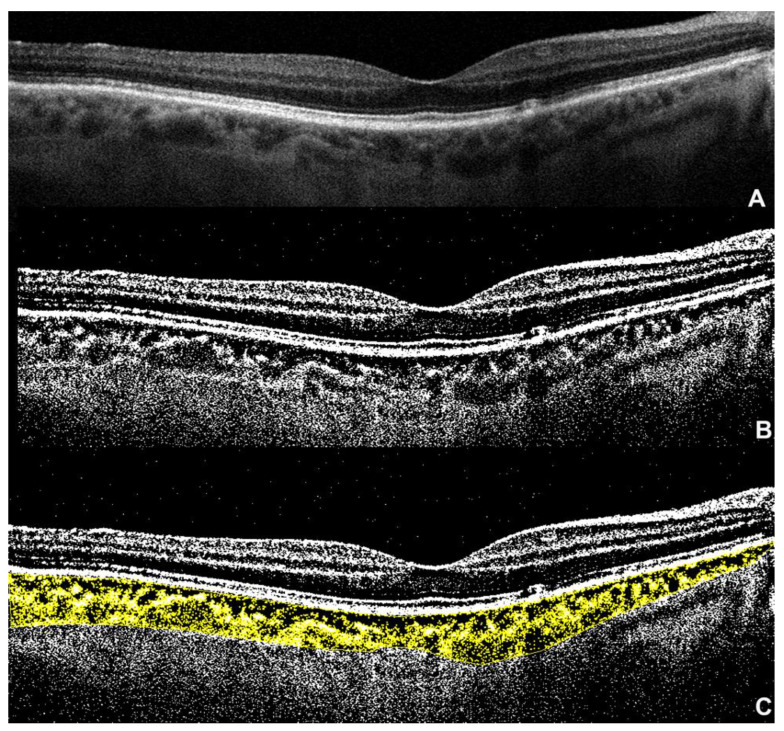
Choroidal vascularity index (CVI): (**A**) spectral-domain optical coherence tomography (SD-OCT) subfoveal B-scan; (**B**) the same image after binarization using Niblack’s autolocal threshold technique; (**C**) the binarized image after applying color threshold into a polygonal selection traced between the retinal pigment epithelium and the choroid–scleral junction; the dark pixels represent the luminal areas, while the remaining light pixels represent the stroma.

**Figure 3 ijms-23-12010-f003:**
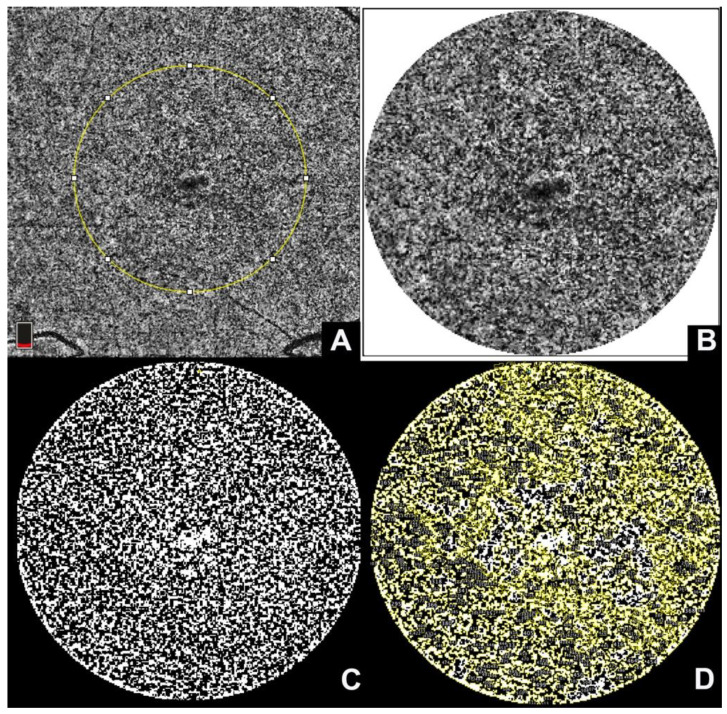
Choriocapillaris flow voids: (**A**) a choriocapillaris slab obtained through optical coherence tomography angiography. The yellow circle represents a region of interest chosen for processing. (**B**) The region of interest is then cropped using a circular selection of 1 mm and (**C**) imported into Fiji software for binarization using autolocal threshold with Phansalkar method; (**D**) the binarized slab is analyzed using the “Analyze Particles” tool that automatically counts the flow deficits.

## Data Availability

Not applicable.

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
