# Peer review of "Choroidal Vasculature Changes in Age-Related Macular Degeneration: From a Molecular to a Clinical Perspective"

_ijms, 2022, doi:10.3390/ijms231912010_

Round 1

Reviewer 1 Report

However, this is an interesting review of modern imaging of choroid and retina, I can not see the link between the molecular biology and pathology with these techniques. Would you please describe more precisely what are the associations between these imaganing techniques and results of molecular examinations , are there any ? You should also include above idea in the limitations and concluisons sections.  

Author Response

Response to Reviewer 1 Comments

Point 1: However, this is an interesting review of modern imaging of choroid and retina, I can not see the link between the molecular biology and pathology with these techniques. Would you please describe more precisely what are the associations between these imaganing techniques and results of molecular examinations , are there any ? You should also include above idea in the limitations and concluisons sections. 

Response 1: We profoundly thank the R1 for the appreciation and the guidance provided. We totally agree on the need to better specify the associations and potentialities of these imaging techniques in understanding the age-related modifications of the choroid. The extraordinary advances in non-invasive imaging techniques have permitted an almost histological resolution of the choroidal layers, making possible the recognition of ultrastructural alterations of the choroid on the different anatomical planes. To address this critical point, we have expanded chapter 5 “choroidal imaging and assessment in vivo” providing an exhaustive paragraph in this regard. Please check p. 6, l.272-288.

Further, following the Reviewer’s suggestions, we have also mentioned the above topic within the limitations and conclusion. Please check p. 12, l. 519-524.

Reviewer 2 Report

I enjoyed reading this review and like to idea of focusing on the choroid in AMD. I find, however, that there was little to review for most parts, which makes this review reasonably underpowered. I would find this work more appropriate for publication if this were a reflection on current literature and perspective for future jobs. While there was some evaluation of the existing literature, I believe a stricter assessment of the results from the refereed literature would have been critical. Can the CVI be used for the clinic yet? There is no reflection on its usability, power to detect changes and the reliability of the data published in the past.

There were several issues with references. I found several that quoted the wrong references. I suggest carefully reevaluating this throughout the text and ensuring that the ref matches the paper they want to quote.

I believe this work could be shaken into a very appropriate form with some extra considerations.

Author Response

Response to Reviewer 2 Comments

Point 1: I enjoyed reading this review and like to idea of focusing on the choroid in AMD. I find, however, that there was little to review for most parts, which makes this review reasonably underpowered. I would find this work more appropriate for publication if this were a reflection on current literature and perspective for future jobs. While there was some evaluation of the existing literature, I believe a stricter assessment of the results from the refereed literature would have been critical. Can the CVI be used for the clinic yet? There is no reflection on its usability, power to detect changes and the reliability of the data published in the past.

I believe this work could be shaken into a very appropriate form with some extra considerations.

Response 1: We are thankful for the reviewer’s comments and suggestions. This narrative review aimed to report an overview of the choroidal role in age-related macular degeneration from a molecular to a clinical perspective. We have described only the commercially available technologies and the validated imaging parameters to offer a practical and current perspective on how to study the choroidal vascular layer in vivo. Since the topic was quite extensive, we have tried to condense the most relevant points. We apologize if this was not clear enough; therefore, we have significantly expanded chapter 5 accordingly. Please check on p. 6, l. 272-288. We have added considerations on the choroidal thickness evaluation on p. 7, l. 308-315 and on p. 8, l. 348-353.

Please note that we have already specified the utility and the application of CVI in several retinal disorders (p. 9, l. 412-417) and the greater sensitivity with less variability compared to the choroidal thickness evaluation (p. 9, l. 419-424). To further expand the description of CVI, including its reproducibility and remarking its utility in clinical practice, we have added the following paragraphs on p. 9, l. 412-413 and p. 9-10, l. 424-436. We have also added the potential limitation of the technique on p. 10, l. 436-442. Please also check the limitations and conclusions section on p. 12, l. 519-524.

Point 2: There were several issues with references. I found several that quoted the wrong references. I suggest carefully reevaluating this throughout the text and ensuring that the ref matches the paper they want to quote.

Response 2: We thank the Reviewer for raising this point, as we agree that the bibliographic references' accuracy is a priority. To avoid mistakes in the bibliographic referencing, we have used the commercially available and licensed software EndNote (version X 9.3.3). As we cannot exclude having made mistakes by erroneously selecting the wrong reference, we have rechecked all the references included in the present manuscript version.

Round 2

Reviewer 1 Report

I am satisfied with the changes of this manuscript

Reviewer 2 Report

The authors addressed the issues I raised and I have no further comments.